# Structural Features of Nucleoproteins from the Recently Discovered *Orthonairovirus songlingense* and *Norwavirus beijiense*

**DOI:** 10.3390/ijms26157445

**Published:** 2025-08-01

**Authors:** Alexey O. Yanshin, Daria I. Ivkina, Vitaliy Yu. Tuyrin, Irina A. Osinkina, Anton E. Tishin, Sergei E. Olkin, Egor O. Ukladov, Nikita S. Radchenko, Sergey G. Arkhipov, Yury L. Ryzhykau, Na Li, Alexander P. Agafonov, Ilnaz R. Imatdinov, Anastasia V. Gladysheva

**Affiliations:** 1State Research Center of Virology and Biotechnology “Vector”, 630559 Kol’tsovo, Russia; yanshin_ao@vector.nsc.ru (A.O.Y.); ivkina_di@vector.nsc.ru (D.I.I.); tyurin_vyu@vector.nsc.ru (V.Y.T.); osinkina_ia@vector.nsc.ru (I.A.O.); tishin_ae@vector.nsc.ru (A.E.T.); olkin@vector.nsc.ru (S.E.O.); radchenko_ns@vector.nsc.ru (N.S.R.); agafonov@vector.nsc.ru (A.P.A.); imatdinov_ir@vector.nsc.ru (I.R.I.); 2Physics Department, Novosibirsk State University, 630090 Novosibirsk, Russia; 3Scientific Educational Center Institute of Chemical Technology, Novosibirsk State University, 630090 Novosibirsk, Russia; e.ukladov@g.nsu.ru (E.O.U.); arksergey@gmail.com (S.G.A.); 4SRF “SKIF”, 630559 Kol’tsovo, Russia; 5Research Center for Molecular Mechanisms of Aging and Age-Related Diseases, Moscow Institute of Physics and Technology, 141701 Dolgoprudny, Russia; rizhikov1@gmail.com; 6National Facility for Protein Science Shanghai, Shanghai Advanced Research Institute, No. 99, Haike Road, Shanghai 201210, China; lina02@sari.ac.cn

**Keywords:** *Ixodid tick*, tick-borne infection, RNA viruses, viral proteins, nucleocapsid, ribonucleoprotein complex, protein structure, small-angle X-ray scattering, AlphaFold

## Abstract

The recent discovery of *Orthonairovirus songlingense* (SGLV) and *Norwavirus beijiense* (BJNV) in China has raised significant concern due to their potential to cause severe human disease. However, little is known about the structural features and function of their nucleoproteins, which play a key role in the viral life cycle. By combining small-angle X-ray scattering (SAXS) data and AlphaFold 3 simulations, we reconstructed the BJNV and SGLV nucleoprotein structures for the first time. The SGLV and BJNV nucleoproteins have structures that are broadly similar to those of *Orthonairovirus haemorrhagiae* (CCHFV) nucleoproteins despite low sequence similarity. Based on structural analysis, several residues located in the positively charged region of BJNV and SGLV nucleoproteins have been indicated to be important for viral RNA binding. A positively charged RNA-binding crevice runs along the interior of the SGLV and BJNV ribonucleoprotein complex (RNP), shielding the viral RNA. Despite the high structural similarity between SGLV and BJNV nucleoprotein monomers, their RNPs adopt distinct conformations. These findings provide important insights into the molecular mechanisms of viral genome packaging and replication in these emerging pathogens. Also, our work demonstrates that experimental SAXS data can validate and improve predicted AlphaFold 3 structures to reflect their solution structure and also provides the first low-resolution structures of the BJNV and SGLV nucleoproteins for the future development of POC tests, vaccines, and antiviral drugs.

## 1. Introduction

Emerging pathogenic tick-borne viruses capable of infecting humans and animals have attracted much attention due to the increasing incidence of tick-borne viral infections and their significant impact on human health [1]. Ticks, acting as carriers, play a crucial role in the transmission of these viruses to humans and animals. Ixodid ticks are among the most versatile carriers due to their ability to transmit several types of pathogens to humans and animals at once, including bacteria, protozoa, and viruses [2]. In recent years, NGS has been used to identify emerging and re-emerging putative human infections caused by tick-borne viruses belonging to the *Nairoviridae* family that are phylogenetically distinct from previously reported human pathogenic viruses. These include *Orthonairovirus tomdiense*, Sulina virus (*Orthonairovirus sulinaense*), Tacheng tick virus 1 (*Orthonairovirus tachengense*), Songling virus (*Orthonairovirus songlingense*, SGLV), Yezo virus (*Orthonairovirus yezoense*, YEZV), Beiji nairovirus (*Norwavirus beijiense*, BJNV), and Iwanai valley virus [3,4,5,6,7,8,9].

SGLV was first discovered in 42 patients from China in regions bordering Russia between 2017 and 2018 with symptoms including headache, fever, depression, fatigue, and dizziness [6]. BJNV was first identified in 67 patients from Inner Mongolia, China, in 2017–2018 with similar symptoms. In one case, the disease caused by BJNV resulted in the death of the patient [8]. In addition, SGLV and BJNV genetic material was found in ticks: *Ixodes persulcatus* (SGLV, BJNV), *Ixodes crenulatus* (SGLV, BJNV), *Demacentor silvarum* (BJNV), *Demacentor nuttalli* (SGLV, BJNV), *Haemaphysalis conicinna* (SGLV, BJNV), *Haemaphysalis longicornis* (SGLV, BJNV), *Rhipicephalus sanguineus* (BJNV), and *Rhipicephalus microplus* (BJNV) [10,11,12,13].

SGLV belongs to the *Orthonairovirus* genus of the *Nairoviridae* family. The SGLV genome is represented by tri-segmented, single-stranded RNA of negative polarity (ssRNA(−)). The S segment is approximately 1.5 kb long and encodes the 487 aa viral nucleoprotein (N). The M segment is approximately 4 kb long and encodes the 1355 aa viral glycoprotein precursor (GPC). The L segment is approximately 12 kb long and encodes the 3951 aa large protein with its RNA-dependent RNA polymerase (RdRp), which is essential for viral replication. BJNV belongs to the *Norwavirus* genus of the *Nairoviridae* family. Unlike SGLV, the BJNV genome is represented by a bi-segmented ssRNA(−) of approximately 18.6 kb. The S segment is approximately 3.7 kb long and encodes the 552 aa N; the L segment is approximately 14.9 kb long and encodes the 4811 aa large protein with its RdRp. The *Nairoviridae* family includes several important tick-borne pathogens that cause serious disease in humans and animals. The most prominent member of this genus is the Crimean–Congo hemorrhagic fever virus (*Orthonairovirus haemorrhagiae*, CCHFV), which causes severe hemorrhagic fever in humans with reported mortality rates of 5% to 30% [14].

Both viruses encode N proteins in their genome structure, which play an important role in encapsulating the viral genome to form a ribonucleoprotein complex (RNP). RNP is necessary for protecting the viral genome from degradation, folding of viral RNA, and replication and transcription of the viral genome. However, it has been shown that the role of the N protein of ssRNA(−) viruses is much broader and includes participation in the host’s innate immune responses to viral infection [15].

This work is devoted to the first decoding of the SGLV N and BJNV N tertiary structures and the prediction of their probable functions using modern methods of synchrotron small-angle X-ray scattering and structural bioinformatics. This information has not been described in the literature so far and is fundamentally important for the further development of preventive measures against viral threats associated with recently discovered SGLV and BJNV, which have already caused serious infectious diseases in humans.

## 2. Results

In an effort to elucidate the structural features of full-length SGLV N and BJNV N, we investigated the native-state conformations of SGLV N and BJNV N through an integrated approach combining SAXS across multiple protein concentrations and tertiary structure modeling with AlphaFold 3.

### 2.1. AlphaFold 3 Tertiary Structure Models of SGLV N and BJNV N

Tertiary structure models of SGLV N with pLDDT = 81 and BJNV N with pLDDT = 73 were generated using AlphaFold 3 (Appendix A). SGLV N and BJNV N were found to have the highest topological similarity to CCHFV N with TM-score 0.87 (PDB ID: 4AQF) and TM-score 0.80 (PDB ID: 6Z0O), respectively, despite extremely low amino acid sequence identity (Table 1). While when comparing the tertiary structures of SGLV N, BJNV N, and the N crystal structures of other closely related viruses, significantly lower similarity coefficients were observed. Also, the tertiary structure model of the SGLV N differs from those of the BJNV, with TM-score 0.78 and RMSD 4.34 Å (Table 1).

The SGLV N and BJNV N have a two-domain racket-shaped structure typical for orthonairoviruses nucleoproteins, consisting of a head domain and a stalk domain. Both SGLV N and BJNV N domains mainly consist of α-helices (Figure 1). In addition, the BJNV N has an extended unstructured N-terminal tail of 35 amino acids, which is not observed in the SGLV N.

The structure of SGLV N and BJNV N head domains are more conserved than the stalk domains. The SGLV N and the BJNV N structures show high flexibility of the stalk domain with TM-score 0.59 and RMSD 3.50 Å (Table 1).

While the SGLV N and BJNV N head domains align very closely, the SGLV N and BJNV N stalk domains adopt radically different positions. Significant conformational differences are exhibited in the orientation between the head and the stalk domains. The SGLV N stalk domain contains four alpha helices (α9–α12) with several tight turns, while the BJNV N stalk domain has four α-helices (α9–α12) and one beta sheet with two β-strands (β2 and β3) and a tight turn. It is noteworthy that the tertiary structure of the CCHFV does not contain a beta sheet (Figure 1). A two β-strand motif was observed in the stalk domain of Kupe virus N (PDP ID: 4XZC) (Appendix A).

However, SGLV N and BJNV N AlphaFold 3 models may have some regions that are incorrectly predicted due to the lack of structural information from homologous viral proteins in the PDB database and the limited ability of the AlphaFold 3 to predict viral proteins. In order to test the suitability of the SGLV N and BJNV N AlphaFold 3 models for structural analysis, we produced recombinant SGLV N and BJNV N and measured them using the SAXS method under native conditions (20 mM Tris-HCl and 150 mM NaCl, pH 7.5; see Section 4. Materials and Methods).

### 2.2. Recombinant SGLV N and BJNV N

We successfully obtained highly purified (>95% by SDS-PAGE) full-length recombinant SGLV N (487 aa, pI 7.66) and BJNV N (556 aa, pI 7.73) in a soluble form, without affinity tags and fusion proteins, using a prokaryotic cell system. SGLV N eluted at a peak of 14.9 mL on a Superdex^®^200 Increase 10/300 GL column, corresponding to the SGLV N monomeric form with a molecular weight of 54.5 kDa, and BJNV N eluted at a peak of 16.1 mL, corresponding to the BJNV N monomeric form with a molecular weight of 61.4 kDa (Figure 2a,c and Appendix A). The A260/A280 ratio of approximately 0.5 indicates that the SGLV N and BJNV N were successfully stripped of endogenous nucleic acids from the expression host. SEC and DLS analysis showed that SGLV N and BJNV N mainly exist in a monomeric form with DNase/RNase treatment similar to CCHFV N [17]. It is noteworthy that the SGLV N size was 6.02 nm ± 1.8, while the BJNV N size was 8.47 nm ± 2.05, which is significantly larger than that of the SGLV N (Figure 2b). This pronounced difference in hydrodynamic size—with BJNV N appearing significantly larger than SGLV N by DLS—is surprising, given that the molecular weight of BJNV N (61.4 kDa) is only slightly higher than that of SGLV N (54.5 kDa). A likely explanation is the presence of an extended, intrinsically disordered region at the C-terminus of BJNV N (residues 516–551), which can substantially increase the apparent hydrodynamic radius. This hypothesis is supported by hydrodynamic modeling based on AlphaFold 3 structures using the HullRad server, which predicts diameters of 6.30 nm and 7.62 nm for SGLV N and BJNV N, respectively—closely matching the experimental DLS data (Appendix A and Figure 2b).

### 2.3. SAXS Data of SGLV N and BJNV N

Overall structural parameters of SGLV N and BJNV N, such as the molecular weight (MW), the radius of gyration of the molecule (Rg), the maximum linear dimension (Dmax), and the distance distribution function (P(r)), were calculated using the PRIMUS of the ATSAS v4.0 package. The SGLV N and BJNV N ab initio models were reconstructed in 3D at low resolution using DAMMIF by ensuring P1 symmetry. Then, averaging of the SGLV N and BJNV N ab initio models (15 for each protein) was performed in DAMAVER. The SGLV N and BJNV N structures modeled in the AlphaFold 3 were fitted to the SGLV N and BJNV N ab initio models from SAXS data using the ChimeraX v1.15rc (Appendix A).

SGLV N and BJNV N show an increase in the molecule’s structures in solution (slightly larger Rg and Dmax) compared to the structures predicted by the AlphaFold 3 and DLS data. This is especially pronounced for the stalk domain, reflecting the SGLV N and BJNV N conformational dynamics in solution. The shape of the SAXS curve does not change depending on the SGLV N and BJNV N concentration in the solution (Figure 3a,d). Rg increases slightly by ~0.2 nm for SGLV N and does not change for BJNV N with increasing concentration (Table 2). P(r) exhibits one maximum. The position of the P(r) maximum does not change depending on the SGLV N and BJNV concentration (Figure 3c,f). The absence of significant upward or downward slope in the intensity profiles and the linearity of the Guinier plots for the three concentrations of SGLV N and BJNV N indicate the absence of protein aggregation or significant interparticle effect in the concentration range studied. A slight increase in Rg, Dmax, and V(Porod) for SGLV N at the highest concentration (9.8 mg/mL) suggests minor concentration-dependent oligomerization. However, this effect is negligible at 1.01 and 3.15 mg/mL, where scattering patterns are consistent. Thus, data from the lower concentrations were used for further analysis. The Kratky plots provide additional insight: for SGLV N, the peak position closely aligns with the theoretical value expected for a compact globular particle, supporting its well-folded, compact conformation (Figure 3b and Appendix A). The BJNV N Kratky profile shows a small shift from this reference point, which is consistent with the presence of an intrinsically disordered C-terminal segment (Figure 3e and Appendix A). The difference between the expected and experimentally obtained SGLV N and BJNV N molecular weights may arise due to error of theoretical molecular weight calculations (Table 2).

To test the plausibility of the SAXS models, AlphaFold 3 structural models of the BJNV N and SGLV N and crystal structures of N of viruses having the highest topological similarity were generated and fitted into the corresponding SAXS-derived shells. We compared the SAXS data for SGLV N at 3.15 mg/mL and BJNV N at 3.46 mg/mL with both the crystallographic data of viral N and the in silico predictions from AlphaFold 3. The experimental SAXS curves and theoretical curves were compared by analyzing the associated chi-square (χ^2^) test for the BJNV N and SGLV N (Appendix A). The χ^2^ variance should be equal to 1 for the ideal case. Our results show a χ^2^ = 1.1–1.3 for BJNV N and a χ^2^ = 1.2–1.3 for SGLV N (Table 3). Overall, SAXS measurements indicate fewer compact structures. The differences observed may be due to the fact that the structure in solution may differ slightly from the structure in the crystal and, in particular, reflect multiple conformational spaces in one averaged model. However, given the low resolution of the SAXS method, we conclude that the structures of the SGLV N and BJNV N are well preserved in solution and the AlphaFold 3 structural models of the BJNV N and SGLV N are adequate.

The solved tertiary structures obtained from ab initio modeling based on the SAXS scattering curves are shown in Figure 4. SGLV N and BJNV N tertiary structures have similar elongated shapes and are similar in size. The AlphaFold 3 structure fits well into the ab initio tertiary structure models of SGLV N and BJNV N, confirming that there are no major conformational changes between the structures in solution and confirming the suitability of the AlphaFold 3 tertiary structure models of SGLV N and BJNV N for further structural and functional analysis.

### 2.4. Interaction of SGLV N and BJNV N with Their ssRNA(−)

Three positively charged regions are found in the SGLV N and BJNV N structures (Figure 5a,c). Two regions, representing a deep positively charged crevice (BJNV N crevice: Lys49, Arg53, Arg185, Lys363, Lys364, Lys365, His392, His401, Arg402, His472, His475; SGLV N crevice: Arg54, Arg175, Arg383, Lys342, Lys344, His452, His455) and a platform (BJNV N platform 1: Arg62, Lys88, Lys136, His489; SGLV N platform 1: Arg87, Arg88, Lys95, Lys147), are located in the head domains, and a third positively charged region (BJNV N platform 2: Lys191, Lys193, Lys258; SGLV N platform 2: Lys187, His192, Lys216, Arg219) is located in the stalk domains. Structural alignment revealed that the SGLV N and BJNV N RNA-binding regions correspond to those within CCHFV N (Appendix A). The essential residues identified in these regions are well conserved in N proteins of other members of the *Nairoviridae* family and thus suggest a role in RNA binding [18].

Modeling of the structure of SGLV N and BJNV N monomers in complex with fragments of their ssRNA(−) of the S segment (SGLV RNA: 3′-UUUGAAUAUGGGGUGUGCA-5′ and BJNV RNA: 3′-GGGUUAGAUAUCUUCGAGA-5′) showed that SGLV RNA and BJNV RNA interact with N proteins in the crevice and platform 2, which supports the hypothesis that these positively charged regions are RNA-binding (Figure 5b,d). SGLV RNA and BJNV RNA literally penetrate SGLV N and BJNV N in the crevice, forming extensive hydrogen bonds, and extend out through platform 2, leaving their ends available for interaction with the next SGLV N and BJNV N monomers. In the crevice of SGLV N and BJNV N there are two extended RNA-binding regions. Both RNA-binding regions of the crevice in SGLV N and BJNV N have a contact distance with the RNA of less than 3.2 Å (Appendix A).

Interestingly, when modeling BJNV N and SGLV N in complex with another ssRNA(−) fragment of the BJNV S segment, for example, 3′-CUCACGGAUUACUGGUUGA-5′, we observed movement of the BJNV N and SGLV N stalk domain toward the formation of a more compact BJNV N tertiary structure (Appendix A). Notably, platform 1 of both SGLV N and BJNV N remains uninvolved in RNA-binding in these simulations. It is possible that this region becomes functional upon oligomerization of viral N proteins or upon binding of large RNA, or that double-stranded RNA is required for this region to function.

We attempted to reproduce the oligomeric state of SGLV N and BJNV N in complex with their ssRNA(−) using AlphaFold 3. SGLV RNPs and BJNV RNPs exhibit distinct oligomeric structures (Figure 6 and Appendix A). The trimers, tetramers, pentamers, and hexamers of SGLV RNPs and BJNV RNPs adopt ring-shaped configurations. The oligomeric structures of SGLV RNPs form a large central cavity (~50–60 Å in diameter), whereas BJNV N RNPs assemble into more compact structures with multiple smaller cavities within the ring. The heptamers of SGLV RNPs and BJNV RNPs adopt a helical conformation, consisting of repeating six-monomer units, with the seventh monomer coiled into a helical turn (Figure 6a,b). The SGLV N-SGLV N interface comprises domain contacts between tight turns within α9–α10 and α11–α12 of the stalk domain and the region within α15–α16 of the head domain (Appendix A). The BJNV N-BJNV N interface comprises domain contacts between α9-TT-β2 of the stalk domain and α2 and α14–α15 of the head domain (Figure 6c,d, Appendix A). A positively charged RNA-binding crevice runs along the interior of SGLV RNP and BJNV RNP, shielding the viral ssRNA(−).

## 3. Discussion

In this study, we present the first structural investigation of BJNV N and SGLV N through an integrative approach combining SAXS experiments with AlphaFold 3 computational modeling.

The BJNV N and SGLV N monomers have a linear molecular dimension of approximately 9–10 nm (~R_g_ = 2.7 ± 0.1–3.1 ± 0.1) and a molecular weight of 47.6–62.0 kDa that varies slightly depending on the concentration. The BJNV N and SGLV N have a racket-shaped tertiary structure with distinct head and stalk domains that are broadly similar to those of CCHFV N, despite sequence similarities of 22% and 35.6%, respectively. The similarity of the BJNV N and SGLV N tertiary structures with the CCHFV N tertiary structure may indicate the similarity of viral functions and mechanisms of implementation of viral genetic information. CCHFV is a tick-borne orthonairovirus capable of causing hemorrhagic fever in humans, with mortality rates reaching up to 40%. The CCHFV N is widely distributed in the virion, participates in genome encapsidation, RNP formation, viral replication, and transcription, interacts with various cellular proteins and factors, plays an important role in viral pathogenesis, and stimulates innate, humoral, and cellular immune responses [19]. The observed preservation of tertiary structure across divergent ssRNA(−) viruses suggests conservation of fundamental functional mechanisms in viral genome processing.

According to crystal structures, the head domains of the orthonairoviruses N present the most conserved architecture and consist predominantly of α-helices. Interestingly, the head domain residues of BJNV N and SGLV N form a parallel double-β-stranded sheet, similar to the low-homology structures of Erve virus N (PDB ID: 4XZ8) and Kupe virus N (PDB ID: 4XZC). In contrast, the highly homologous head domains of CCHFV N (PDB ID: 4AQF) and Hazara virus N (PDB ID: 4XZE) lack this feature (Appendix A). Despite this structural difference, the head domains of SGLV N and BJNV N closely resemble those of CCHFV N, whereas their stalk domains adopt radically different positions. On the one hand, this may be due to the limitations of the chosen SAXS and AlphaFold 3 methods. On the other hand, this rearrangement may have important consequences that result in different activities in critical SGLV N and BJNV N functions, such as protein oligomerization or viral RNA binding.

Interestingly, bacterially expressed and purified SGLV N and BJNV N mainly exist in a monomeric form, similar to CCHFV N, according to data of DLS, SEC, and SAXS. This is quite different from the situation during the other reported ssRNA(−) viruses, which remained strongly and stably bound to host cell nucleic acids and oligomerized [20,21].

Notably, for SGLV N and BJNV N and the N proteins of homologous viruses, one of the regions of coupling between the head and stalk domains is not constructed for crystal structures due to the lack of interpretable electron density, which also indicates its structural flexibility. This coupling region corresponds to the α6 helix in the Lassa virus N, which adopts an open state upon binding to RNA during the so-called “gating mechanism” [22]. The other remaining region of coupling between these two domains in nucleoproteins is the α12, but this single helix is not sufficient to establish a rigid connection between the head and stalk domains (Appendix A). Thus, the coupling region between these two domains is very flexible in N proteins, which allows for interchangeable conformational changes for some key N functions such as RNP oligomerization or binding to viral RNA.

Next, we investigated the interaction of SGLV N and BJNV N with their ssRNA(−) and the oligomerization of SGLV and BJNV RNPs. It has been previously reported that CCHFV N has different binding modes for viral RNA and undergoes a conformational change in the presence of RNA [23,24]. A similar phenomenon is observed for BJNV N, but not for SGLV N. SGLV N and BJNV N binds RNA differently, possibly due to divergent evolutionary adaptations. This observation highlights a critical structural and functional feature shared by many ssRNA(−) viruses, particularly in the *Bunyavirales* order. Also, both SGLV N and BJNV N form trimers, tetramers, pentamers, and hexamers in ring-shaped arrangements and helical heptamers, typical of many nucleoproteins of ssRNA(−) viruses. However, SGLV RNPs have a larger central cavity, suggesting a more open architecture and the ability to accommodate bulkier RNA segments or allow dynamic rearrangements during viral replication.

Determining the structure of viral proteins is an important step in the development of antiviral drugs. Bioinformatic analysis has repeatedly identified candidate antiviral drugs, thereby expanding our arsenal of means to combat viral infections [25,26,27]. Being highly conserved proteins, the analogues of which are absent in human cells, N proteins are an interesting target for the search for new antiviral drugs. Thus, several substances have been shown to block the work of the nucleoprotein of the influenza A virus [28,29]. A recent study showed that the FDA-approved drug lurazidone suppresses severe fever caused by the thrombocytopenia syndrome virus [30]. The authors showed that lurazidone has an antiviral effect by directly binding to the viral nucleoprotein, thereby disrupting genome replication. These results provide a valuable theoretical basis for the rational development of new antiviral drugs targeting viral nucleoproteins.

The combination of SAXS with high-resolution structural techniques and computer analysis represents a powerful strategy for converting structural knowledge into effective low-molecular-weight inhibitors. This comprehensive approach is particularly promising for combating viruses carried by ticks and mosquitoes, where there is an urgent need for new therapeutic agents. Thus, our work demonstrates that experimental SAXS data can validate and improve predicted AlphaFold 3 structures of recently discovered viruses to reflect their solution structure and can be used in concert to directly analyze functions and conformational changes in nucleoproteins of recently discovered viruses that have extremely low genomic sequence identity compared to annotated viral genomes. Understanding the structure and function of viral nucleoproteins offers advantages in finding new therapeutic targets and also enhances our understanding of viral evolution. The results of this study are fundamentally important for the further development of preventive measures against viral threats associated with recently discovered SGLV and BJNV, which have already caused serious infectious diseases in humans. Further studies of nucleoproteins of poorly understood and novel ssRNA(−) viruses will undoubtedly reveal even more fascinating and diverse aspects.

## 4. Materials and Methods

### 4.1. Synthetic Assembly of DNA Copies of Genes Encoding Viral Nucleoproteins

Synthetic, codon-optimized DNA copies of the genes encoding the SGLV N (1449 bases, GenBank Acc. No. NC_079002, codon-optimized) and BJNV N (1653 bases, GenBank Acc. No. MW315111, codon-optimized) were obtained by sequential annealing of partially complementary oligonucleotides followed by PCR amplification (Appendix A). DNA fragments were amplified using the Q5 High-Fidelity DNA Polymerase kit (NEB, Hitchin, UK). Amplicon analysis was performed at each stage using electrophoresis in 2% agarose gel. Amplicons were purified using the Cleanup Standard kit (Eurogen, Moscow, Russia). The assembled DNA copies of the SGLV N and BJNV N genes were directly cloned into the pJET1.2/blunt plasmid (Thermo Fisher Scientific, Waltham, MA, USA) to create a genetic construct that ensures long-term storage of the synthesized DNA copies. Then, using the “heat shock” method, the pJET1.2/SGLV-N and pJET1.2/BJNV-N genetic constructs, containing DNA copies of the SGLV N and BJNV N genes, were transformed into chemically competent cells of *Escherichia coli* strain NEB Stable (NEB, UK) and plated on Petri dishes with LB agar medium (AppliChem, Darmstadt, Germany) with ampicillin at a concentration of 100 μg/mL as a selective marker, at a temperature of 30 °C for 16–18 h. Untransformed *Escherichia coli* cells of the NEB Stable strain served as a negative control. Recombinant plasmid DNA was isolated using the Plasmid Miniprep Color kit (Eurogen, Russia). The correctness of the nucleotide sequence of the pJET1.2/SGLV-N and pJET1.2/BJNV-N genetic constructs was confirmed by sequencing according to the Sanger method using the BigDye Terminator v3.1 kit (Thermo Fisher Scientific, USA) on an ABI 3500/3500xl (Applied Biosystems, Waltham, MA, USA). The obtained sequence data were processed using SnapGene v.3.2.1 (GSL Biotech LLC, San Diego, CA, USA).

### 4.2. Production of Recombinant Viral Nucleoproteins in Escherichia coli

To obtain genetic constructs that ensure the synthesis of SGLV N and BJNV N, DNA amplification of copies of the SGLV N and BJNV N genes from pJET1.2/SGLV-N and pJET1.2/BJNV-N was performed, followed by treatment of the PCR products with the restriction endonuclease *Mal I* (SibEnzyme, Novosibirsk, Russia). SGLV N and BJNV N were cloned into the pET28b-14xHis (Invitrogen, Waltham, MA, USA) vector with *EcoR I* and *Xho I* restriction sites using the cloning primers (Appendix A). The pET28b-14xHis construct was designed in frame with an N-terminal 14xHis tag followed by a small ubiquitin-like modifier fusion protein and a picornain 3C A28 cleavage site (Figure 7). The resulting genetic constructs were transformed into chemically competent *Escherichia coli* cells of the NEB Stable strain. After 16 h of growth on LB agar medium (100 μg/mL kanamycin) at 30 °C, single colonies were selected and grown for 12 h in liquid LB nutrient medium with a selective antibiotic (100 μg/mL kanamycin) at 30 °C and 220 rpm. The recombinant plasmids encoding SGLV N and BJNV N were transformed into *Escherichia coli* strain KRX and overexpressed as an N-terminal 14xHis tag, a small ubiquitin-like modifier fusion protein, and a picornain 3C A28 proteolysis site. Expression of chimeric recombinant SGLV N and BJNV N transgenes was induced by adding IPTG to a concentration of 1 mM and L-rhamnose to a concentration of 0.1%. Induction was carried out for 20–24 h at 18 °C with constant stirring at 180 rpm in a thermostat with a built-in shaker NB-205QF (BioLabs, Shanghai, China).

### 4.3. Purification of Recombinant Viral Nucleoproteins

Cells were collected by centrifugation at 4500× *g* for 15–30 min at 4 °C and resuspended in lysis buffer (20 mM Tris, 500 mM NaCl, 20 mM imidazole, pH 7.5) with the addition of DNase (Biolabmix, Novosibirsk, Russia) and MgCl_2_ (New England BioLab, Hitchin, UK) in a ratio of 1 μL per 10 mL of cell suspension and a cocktail of protease inhibitors (Trans Gene Biotech, Beijing, China) in a ratio of 20 μL per 1 mL of cell suspension. Resuspended cells were sonicated with 20 kHz and 320W on ice using a b.braun LABSONIC 2000 homogenizer (B.Braun, Melsungen, Germany) and purified by centrifugation. The cleared cell lysate was loaded onto a HisPur Ni-NTA Resin (Thermo Fisher Scientific, USA), pre-equilibrated with the lysis buffer. The bound chimeric nucleoproteins were eluted by stepwise increasing the imidazole concentration to 75, 100, 150, 300, and 500 mM by mixing the flow of buffer A (20 mM Tris, 500 mM NaCl, 20 mM imidazole, pH 7.5) with buffer B (20 mM Tris, 500 mM NaCl, 500 mM imidazole, pH 7.5). The optical density was monitored using the HBBiolab software v.1.0 (Hanbon Sci. & Tech., Huaian, China), guided by the readings of the flow photometer included in the Bio-Lab 100 Chromatography System (Hanbon Sci. & Tech., China). Proteolytic cleavage of the N-terminal tract from chimeric nucleoproteins was performed by treatment with picornaine 3C A28 (SRC VB “Vector” Rospotrebnadzor, Koltsovo, Russia) in a molar ratio of 1:8. Proteolysis was carried out at 4 °C for 12–16 h. The proteolysis products were reapplied to a pre-equilibrated column with HisPur Ni-NTA Resin sorbent (Appendix A). Eluted nucleoproteins were further dialyzed using dialysis membrane with MWCO 10 kDa against 300 volumes of buffer C (20 mM Tris and 150 mM NaCl, pH 7.5), concentrated to 10–12 mg/mL, and purified using Superdex^®^200 Increase 10/300 GL (GE Healthcare, Stockholm, Sweden) size exclusion chromatography (SEC). Sodium dodecyl-sulfate polyacrylamide gel electrophoresis (SDS-PAGE) revealed over 95% purity of the final purified recombinant nucleoproteins. The nucleoproteins were quantified and stored in 100 μL aliquots at −80 °C.

### 4.4. Dynamic Light Scattering (DLS) Measurement

DLS data were collected using a BeNano 180 Zeta Pro (Bettersize, Dandong, China) with a light source wavelength of 671 nm, a fixed scattering angle of 173°, and a temperature of 25 °C. The HullRad server was used for calculating hydrodynamic properties of AlphaFold 3-predicted structures and the theoretical molecular weight of proteins, http://52.14.70.9/Run_hullrad.html (accessed on 20 May 2025) [31].

### 4.5. Small-Angle X-Ray Scattering (SAXS)

The SAXS data of nucleoproteins were collected at beamline BL19U2 of the National Facility for Protein Science Shanghai (NFPS) at the Shanghai Synchrotron Radiation Facility (SSRF, Shanghai, China) [32]. The X-ray beam size on the stage was 0.33 mm (H) × 0.05 mm (V). A two-dimensional Pilatus3 2 M detector (DECTRIS Ltd., Baden, Switzerland) was placed at a sample-to-detector distance of 2.7 m. The scattering vector magnitude range (q = (4π/λ)sinθ, where 2θ is the scattering angle and λ = 0.1033 nm is the wavelength) was 0.07–4.5 nm^−1^. SAXS profiles were obtained for different nucleoprotein concentrations. BJNV N samples were prepared at 1.12, 3.46, and 10.65 mg/mL, and SGLV N samples were prepared at 0.99, 3.15, and 9.8 mg/mL concentrations in the buffer (20 mM Tris-HCl and 150 mM NaCl, pH 7.5). Twenty frames with an integration time of 1.0 s were collected for each nucleoprotein’s concentration. The data were normalized to the intensity of the transmitted beam and radially averaged using the BioXTAS RAW program [33]. Scattering profiles of the buffer alone were measured under identical experimental conditions and subtracted from the corresponding sample profiles to account for background contributions and correct for solvent scattering effects. All subsequent data processing and analysis were performed using the ATSAS software package, version 4.0.1 (https://www.embl-hamburg.de/biosaxs/, accessed on 20 May 2025). Initial steps involved standard procedures, including averaging of the scattering curves [34]. Preliminary assessments of data quality and linearity in the low-angle region were carried out using the PRIMUS [35] module.

Key structural parameters such as the radius of gyration (Rg) and forward scattering intensity I(0) were extracted through Guinier analysis:(1)$$I_{exp}\left(s\right)=I\left(0\right)\exp\left(-\frac{s^{2}R_{g}^{2}}{3}\right),$$
was applied within the regime, satisfying the approximation sRg < 1.3, ensuring the linearity criterion was met [36]. These parameters provide essential insight into the overall size and compactness of the macromolecular species in solution. A Bayesian statistical framework was employed to infer the molecular mass of the nucleoproteins directly from the SAXS data, providing both median estimates and credibility intervals with higher reliability. To further dissect the underlying shape and size distribution, p(r) pair-distance distribution functions were computed via an Indirect Fourier Transformation (IFT) approach implemented in GNOM [37] according to Equation (2):(2)$$p\left(r\right)=\frac{1}{2\pi^{2}}\int_{0}^{\infty }srI\left(s\right)\sin\left(sr\right)ds.$$

This real-space transformation allowed for accurate determination of both Rg and the maximum intraparticle dimension (Dmax). The Rg values presented in Table 2 reflect those obtained through p(r) fitting rather than Guinier extrapolation, offering higher resilience to noise. For ab initio low-resolution structural modeling, we employed DAMMIF [38], which reconstructs particle envelopes using a bead-model approach and optimizes conformation via simulated annealing (SA). The modeling strives to minimize the discrepancy between experimental and calculated scattering intensities using a χ^2^ -like target function:(3)$$\chi^{2}=\frac{1}{N-1}\sum_{j}{\left[\frac{I_{exp}\left(s_{j}\right)-cI_{calc}\left(s_{j}\right)}{\sigma \left(s_{j}\right)}\right]}^{2},$$
where *N* is the number of measured data points, $I_{exp}$ and $I_{calc}$ denote the experimental and computed scattering intensities at momentum transfer s, σ represents the uncertainty at each point, and c is a scale factor optimized during the fit.

To ensure structural consistency and reproducibility, multiple independent ab initio reconstructions were generated and compared. Resulting models were spatially superimposed using SUPCOMB [39], and the most representative solutions were identified using the DAMAVER [40] suite via an averaging and filtering protocol. To validate and extend the analysis, theoretical scattering profiles were also calculated using CRYSOL [41], applied to both available crystal structures of related proteins and predicted structural models obtained from AlphaFold 3. This step enabled us to assess the agreement of the experimental data with both atomic-resolution and computationally derived models, providing deeper insight into potential conformational states in solution.

### 4.6. Tertiary Structure Modeling of Nucleoproteins and Their Analysis

Nucleotide sequences determined in this study are available in the GenBank database. Predictive modeling of nucleoprotein tertiary structures was carried out using AlphaFold 3 (Google DeepMind, London, UK), https://alphafoldserver.com/welcome (accessed on 10 May 2025) [42]. A search for closely related proteins whose spatial structures had already been solved experimentally was performed using NCBI BLAST (version 2.17.0) (National Center for Biotechnology Information Basic Local Alignment Search Tool, https://blast.ncbi.nlm.nih.gov) against the PDB database (https://www.rcsb.org/) using the blastp (protein–protein BLAST) algorithm (accessed on 10 May 2025). Nucleoproteins were screened against known viral domains using the FoldSeek server (Seoul National University, Seoul, South Korea), https://search.foldseek.com/ (accessed on 10 May 2025) [43]. Structural inferences were based on per-residue confidence scores provided via the AlphaFold 3 predicted Local Distance Difference Test (pLDDT) metric, which quantitatively estimates deviations in Cα–Cα interatomic distances between the reference and predicted structural models, with values ranging from 0 to 100. The resulting models were selected based on high-confidence structural prediction outputs, with particular emphasis on pLDDT scoring, and aligned against their respective structural homologs using the TM-align algorithm, implemented via the RCSB Pairwise Structure Alignment Tool (Research Collaboratory for Structural Bioinformatics Protein Data Bank, RCSB PDB, Piscataway, NJ, USA), https://www.rcsb.org/alignment/; accessed 10 May 2025). Structural alignments were evaluated according to two key metrics: root mean square deviation (RMSD) to estimate atomic displacement between aligned backbone atoms and TM-score, a length-independent metric that reflects topological similarity on a scale between 0 and 1, with values above 0.5 generally indicating correct fold prediction and values near 1.0 indicating high structural congruency.

Tertiary structure models of viral proteins were visualized using UCSF ChimeraX v1.15rc (University of California, San Francisco, CA, USA) [16]. Comparative secondary structure alignments of viral proteins were generated with ESPript v3.0 (Institute for the Biology and Chemistry of Proteins, Lyon, France), https://espript.ibcp.fr/ (accessed on 10 May 2025) [44], allowing for the graphical representation of conserved secondary structure motifs across multiple aligned sequences.

## Figures and Tables

**Figure 1 ijms-26-07445-f001:**
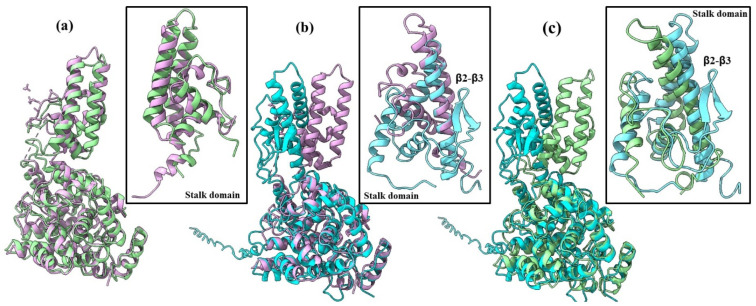
Imposition models of SGLV N and BJNV N tertiary structures: (**a**) SGLV N tertiary structure (green) with CCHFV (PDP ID: 4AQF) (purple); (**b**) BJNV N tertiary structure (blue) with CCHFV (PDP ID: 4AQF) (purple); (**c**) SGLV N tertiary structure (green) with BJNV N tertiary structure (blue). The figure was generated using UCSF ChimeraX v1.15rc [16].

**Figure 2 ijms-26-07445-f002:**
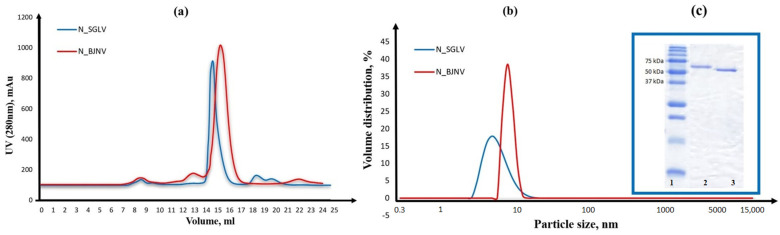
Size exclusion chromatography (**a**), dynamic light scattering (**b**), and SDS-PAGE in 12% gel (**c**) analysis of the recombinant SGLV N and BJNV N. SGLV N is highlighted in blue in (**a**,**b**) and is located on the third lane in (**c**). BJNV N is highlighted in red in (**a**,**b**) and is located on the second lane in (**c**).

**Figure 3 ijms-26-07445-f003:**
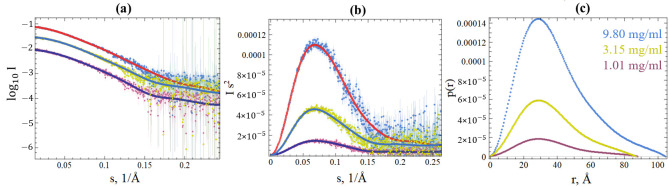
SAXS analysis of full-length SGLV N (**a**–**c**) at concentrations of 9.8 mg/mL (blue), 3.15 mg/mL (yellow), and 1.01 mg/mL (purple) and full-length BJNV N (**d**–**f**) at concentrations of 10.65 mg/mL (pink), 3.46 mg/mL (light blue), and 1.12 mg/mL (green). (**a**) SGLV N scattering experimental data of SGLV N; (**b**) Kratky plot of SGLV N; (**c**) distance distribution function p(r) of SGLV N; (**d**) BJNV N scattering experimental data; (**e**) Kratky plot of BJNV N; (**f**) distance distribution function p(r) of BJNV N. Regularized SAXS curves obtained in GNOM are shown in different colors together with corresponding experimental data.

**Figure 4 ijms-26-07445-f004:**
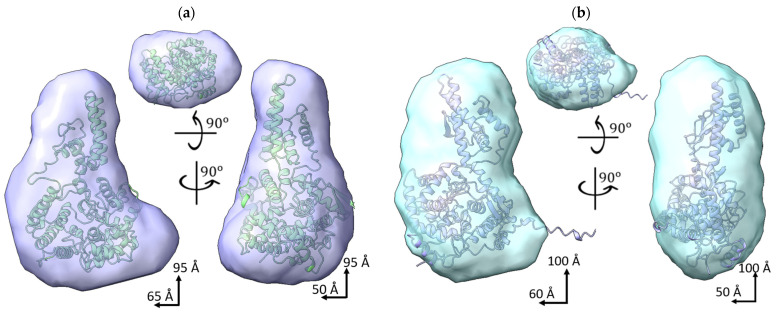
Ab initio tertiary structure models of SGLV N (**a**) and BJNV N (**b**). The AlphaFold model is displayed as ribbon diagrams. The random dummy residues of the average DAMAVER models are shown as a surface.

**Figure 5 ijms-26-07445-f005:**
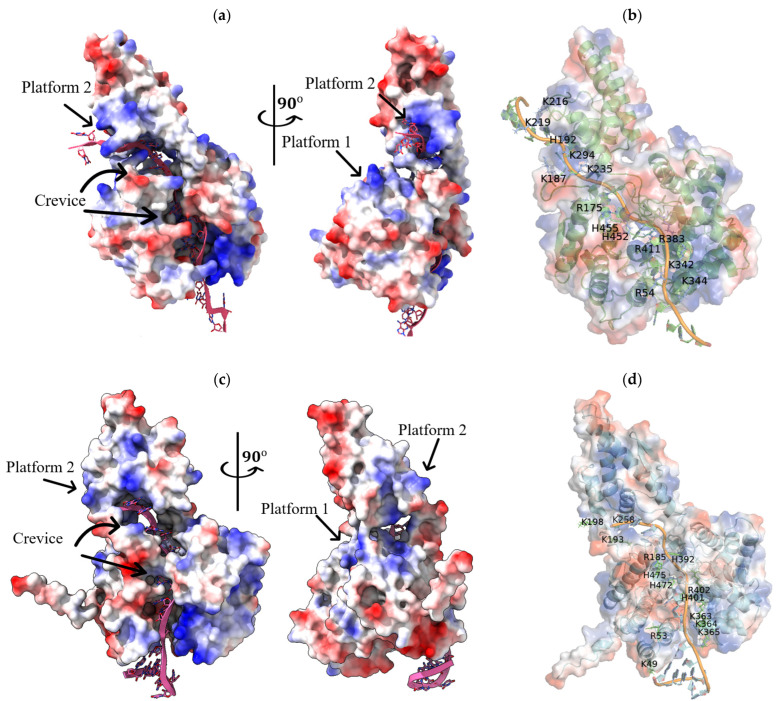
Spatial structures of SGLV N and BJNV N monomers with their RNA fragments of the S genome segment. Electrostatic surface potential of SGLV N (**a**) and BJNV N (**c**). Spatial structures of the SGLV N (purple) with RNA (green) (**b**) and BJNV N (blue) with RNA (orange) (**d**) in ribbon presentation. The positive surface potential is colored blue, and the negative surface potential is colored red. Key amino acids involved in interaction with RNA are highlighted.

**Figure 6 ijms-26-07445-f006:**
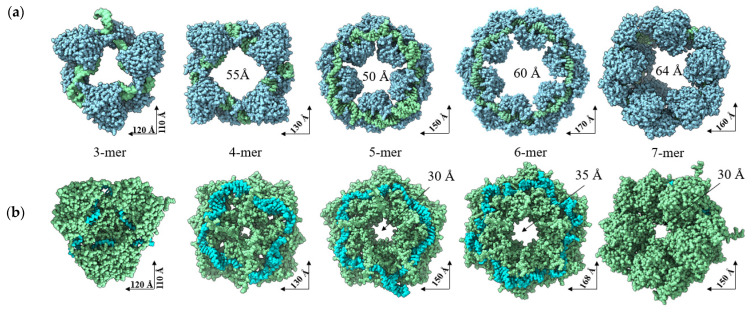
AlphaFold 3 structural models of viral RNP complexes: (**a**,**d**) SGLV RNP (top and side views); (**b**,**c**) BJNV RNP (top and side views). In SGLV RNPs, N monomers are shown in blue, and the RNA is colored green. In BJNV RNPs, N monomers are shown in green, and the RNA is colored cyan. Amino acids associated with the interaction of the BJNV N head domain with ssRNA(−) in the crevice are shown in the orange box and colored in pale pink. Amino acids associated with the interaction of the BJNV N stalk domain with ssRNA(−) in platform 2 are shown in the orange box and colored in pale yellow. Amino acids associated with the BJNV N-BJNV N interaction are shown in the blue box. Amino acids associated with the interaction of the SGLV N head domain with ssRNA(−) in the crevice are shown in the green box and colored in green. Amino acids associated with the interaction of the SGLV N stalk domain with ssRNA(−) in platform 2 are shown in the green box and colored in yellow. Amino acids associated with the SGLV N-SGLV N interaction are shown in the purple box.

**Figure 7 ijms-26-07445-f007:**
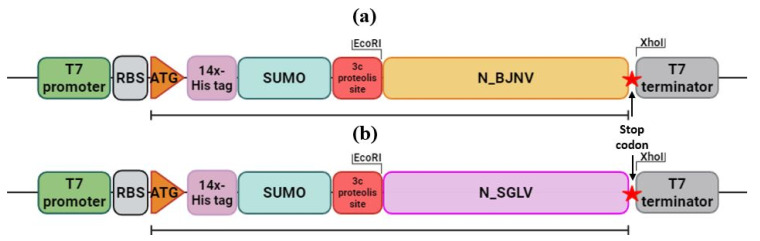
Linear map of BJNV N (**a**) and SGLV N (**b**) expression cassettes.

**Table 1 ijms-26-07445-t001:** Comparison of amino acid sequences and tertiary structures of SGLV N and BJNV N with N proteins of viruses having the highest topological similarity.

PDB ID	Name of Virus	TM-Score	RMSD, Å	Aligned Residues, a.a.	Amino Acid Sequence Identity (Blastp) ^1^, %	Amino Acid Sequence Identity (RSCB) ^2^, %
SGLV N (full-length)						
-	BJNV	0.78	4.34	365	25.0	23.0
4AQF_A	CCHFV	0.87	2.53	433	35.6	34.0
5A97_C	Hazara virus	0.83	3.52	489	32.8	32.0
4AQF_C	CCHFV	0.82	3.27	413	35.4	34.0
4XZE_A	Hazara virus	0.81	3.38	381	32.8	32.0
4XZC_A	Kupe virus	0.78	3.63	367	33.9	28.1
3U3I_A	CCHFV	0.75	2.30	365	36.0	38.0
BJNV N (full-length)						
-	SGLV	0.78	4.34	365	25.0	23.0
6Z0O_B	CCHFV	0.80	3.13	417	22.0	22.0
4AQF_B	CCHFV	0.77	3.48	390	24.0	22.0
4XZC_B	Kupe virus	0.76	3.40	389	23.3	22.0
5A97_A	Hazara virus	0.74	3.98	369	27.6	28.0
4XZA_A	Erve virus	0.64	2.53	337	22.0	25.0
SGLV N head domain						
-	BJNV	0.90	2.15	348	27.0	27.0
4AQF_A	CCHFV	0.94	1.40	351	39.0	40.0
5A97_C	Hazara virus	0.93	1.72	352	36.0	37.0
4AQF_C	CCHFV	0.94	1.46	345	38.0	40.0
4XZE_A	Hazara virus	0.95	1.28	355	35.0	38.0
4XZC_A	Kupe virus	0.95	1.37	355	37.0	39.0
BJNV N head domain						
6Z0O_B	CCHFV	0.90	2.26	342	24.0	26.0
4AQF_B	CCHFV	0.88	2.40	336	25.0	27.0
4XZC_B	Kupe virus	0.89	2.30	339	24.0	26.0
5A97_A	Hazara virus	0.84	2.31	340	27.0	29.0
4XZA_A	Erve virus	0.89	2.57	337	23.0	24.0
SGLV N stalk domain						
-	BJNV	0.55	3.00	85	18.0	18.0
4AQF_A	CCHFV	0.69	2.83	94	17.0	15.0
5A97_C	Hazara virus	0.69	2.68	97	21.0	18.0
4AQF_C	CCHFV	0.73	2.80	99	17.0	16.0
4XZE_A	Hazara virus	0.70	2.85	95	22.0	17.0
4XZC_A	Kupe virus	0.70	2.72	92	21.0	15.0
BJNV N stalk domain						
6Z0O_B	CCHFV	0.56	3.2	82	16.0	14.0
4AQF_B	CCHFV	0.57	3.2	81	15.0	13.0
4XZC_B	Kupe virus	0.58	2.9	81	13.0	11.0
5A97_A	Hazara virus	0.60	3.0	83	15.0	13.0

^1^ Amino acid sequence identity was calculated in the protein–protein BLAST (blastp). ^2^ Amino acid sequence identity was calculated relative to the aligned residues in the tertiary structure.

**Table 2 ijms-26-07445-t002:** Overall structural parameters of SGLV N and BJNV N obtained from SAXS data.

Sample	Concentration, mg/mL	$R_{g}$, nm *	$D_{max}$, nm *	V(Porod), $A^{3}$ *	MW (exp), kDa *	MW (Theory), kDa *
SGLV N	9.8	3.1 ± 0.1	10.3	$103\times 10^{3}$	53.1 [93% CI: 47.2, 57.5]	54.5
	3.15	2.8 ± 0.1	8.7	$86\times 10^{3}$	52.0 [94% CI: 49.2, 55.0]	54.5
	1.01	2.8 ± 0.1	8.8	$81\times 10^{3}$	47.6 [95% CI: 46.2, 51.5]	54.5
BJNV N	10.7	3.2 ± 0.1	9.7	$110\times 10^{3}$	58.1 [91% CI: 53.8, 61.6]	61.4
	3.46	3.0 ± 0.1	9.8	$110\times 10^{3}$	53.1 [91% CI: 50.3, 56.2]	61.4
	1.12	3.2 ± 0.1	9.9	$114\times 10^{3}$	53.2 [91% CI: 53.0, 60.2]	61.4

* V(Porod) is the particle volume, MW (exp) is the experimental molecular weight, and MW (theory) is the theoretical molecular weight calculated according to the SGLV N and BJNV N amino acid sequences.

**Table 3 ijms-26-07445-t003:** Comparison of SGLV N and BJNV N SAXS data with crystal structures of N of viruses having the highest topological similarity and AlphaFold 3 structural models of SGLV N and BJNV N.

Model	$R_{g}$, nm *	$D_{max}$(Envelope), nm	V (Envelope), $A^{3}$	MW, kDa	χ^2^
CCHFV N, PDB ID: 4AQF	2.6	9.1	$81\times 10^{3}$	52.4	BJNV N: 1.2
					SGLV N: 1.2
Hazara virus N, PDB ID: 5A97	2.5	8.8	$82\times 10^{3}$	53.1	BJNV N: 1.3
					SGLV N: 1.3
BJNV N, AlphaFold 3 model	3.0	9.6	$92\times 10^{3}$	61.4	1.1
SGLV N, AlphaFold 3 model	2.5	9.1	$84\times 10^{3}$	54.5	1.3

* R_g_ calculated from the slope of net intensity curve generated by CRYSOL.

## Data Availability

The original contributions presented in this study are included in the article/Appendix A; further inquiries can be directed to the corresponding author.

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
