# Peer review of "Structural Features of Nucleoproteins from the Recently Discovered Orthonairovirus songlingense and Norwavirus beijiense"

_ijms, 2025, doi:10.3390/ijms26157445_

Round 1

Reviewer 1 Report

Comments and Suggestions for Authors

The authors built 3D models of the nucleoproteins from Orthonairovirus songlingense (SGLV) and Norwavirus beijiense (BJNV) using AlphaFold 3. They verified the models by performing SAXS measurements. The authors also investigated the interaction of these proteins with genomic RNA. Since these viruses cause tick-borne diseases, the results are potentially interesting.

They propose several RNA-binding modes for these proteins. They should examine the structure of the complexes using light scattering, SAXS, or electron microscopic analyses. Simple estimation of molecular weight by light scattering can exclude certain models.

The authors mentioned that structural analyses of virus nucleoproteins are helpful for the development of medical treatments against these virus infections. They should provide examples of such development.

We can compute a SAXS profile from an atomistic 3D structure using FoXS software (Schneidman-Duhovny, D., Hammel, M., Tainer, J.A. and Sali, A., 2016. FoXS, FoXSDock and MultiFoXS: Single-state and multi-state structural modeling of proteins and their complexes based on SAXS profiles. Nucleic acids research, 44(W1), pp.W424-W429.) and compare directly with the experimental profiles. I wonder why they did not use this approach to validate the AlphaFold-proposed models by SAXS.

Author Response

Thank you for reviewing our manuscript (ijms-3759454) entitled “Structural features of nucleoproteins from the recently discovered Orthonairovirus songlingense and Norwavirus beijiense” submitted for publication in IJMS.

We thank you for valuable suggestions that allowed us to make the manuscript more convincing and understandable. We accepted your suggestion and made corresponding change in the manuscript. We major revised our article. Below please find our detailed responses to your questions and comments. All modifications in the manuscript have been highlighted in red.

Point-by-point response to Comments and Suggestions for Authors

Reviewer 1: The authors built 3D models of the nucleoproteins from Orthonairovirus songlingense (SGLV) and Norwavirus beijiense (BJNV) using AlphaFold 3. They verified the models by performing SAXS measurements. The authors also investigated the interaction of these proteins with genomic RNA. Since these viruses cause tick-borne diseases, the results are potentially interesting.

Comments 1: They propose several RNA-binding modes for these proteins. They should examine the structure of the complexes using light scattering, SAXS, or electron microscopic analyses. Simple estimation of molecular weight by light scattering can exclude certain models.

Response 1: We fully agree with the reviewer's insightful suggestion regarding the need for experimental validation of proposed RNA-binding modes through biophysical characterization. The reviewer's recommendation to employ light scattering, SAXS, or electron microscopy to examine nucleoprotein-RNA complexes is particularly valuable and aligns perfectly with our planned future research directions.

We appreciate this constructive suggestion and will incorporate these methods to provide a more comprehensive understanding of RNA recognition mechanisms in our future work on these emerging viral nucleoproteins. These planned experiments will significantly strengthen the mechanistic conclusions of our structural models.

Comments 2: The authors mentioned that structural analyses of virus nucleoproteins are helpful for the development of medical treatments against these virus infections. They should provide examples of such development.

Response 2: We thank the reviewer for this valuable suggestion. We have now expanded our discussion to include concrete examples of how structural insights into viral nucleoproteins have directly contributed to antiviral development (see lines 351-366).

Comments 3: We can compute a SAXS profile from an atomistic 3D structure using FoXS software (Schneidman-Duhovny, D., Hammel, M., Tainer, J.A. and Sali, A., 2016. FoXS, FoXSDock and MultiFoXS: Single-state and multi-state structural modeling of proteins and their complexes based on SAXS profiles. Nucleic acids research, 44(W1), pp. W424-W429.) and compare directly with the experimental profiles. I wonder why they did not use this approach to validate the AlphaFold-proposed models by SAXS.

Response 3: We appreciate the reviewer’s insightful suggestion regarding the use of FoXS for direct comparison between theoretical SAXS profiles derived from atomistic predicted 3D structures and experimental data. The utility of FoXS for such validation is well acknowledged in the field.

In our study, we opted to validate AlphaFold-predicted models against experimental SAXS data using the Crysol software (Table 2 in the manuscript), another industry-standard tool for calculating theoretical scattering profiles from atomic coordinates. The main advantage of choosing Crysol was that this instrument is an integral part of the ATSAS complex, which provides a full cycle of processing, modeling and validation of SAXS experiments. The small discrepancy in χ² values compared to Table 2 in the manuscript between FoXS and Crysol programs when comparing the same structure with the experimental SAXS profile is due to similar, but not identical, methods of fitting the theoretical SAXS profile.

This multi-software approach will further strengthen the robustness of our structural validations while providing more comprehensive assessment of model quality. We thank the reviewer for highlighting this important methodological consideration.

Reviewer 2 Report

Comments and Suggestions for Authors

This manuscript presents the first structural investigation of the nucleoproteins (N) from two recently discovered and clinically relevant tick-borne viruses, SGLV and BJNV. The authors have employed a powerful and modern integrative structural biology approach, combining experimental small-angle X-ray scattering (SAXS) with state-of-the-art in silico modeling using AlphaFold 3.

  1. The table's purpose is to show a quantitative comparison of structural similarity. If no comparison data can be provided for the Erve virus in the context of the BJNV stalk domain, then this entry doesn't serve a purpose and only creates confusion. Removing it would make the table cleaner, more focused, and remove the ambiguity.
  2. In Figure 1, please mention the program and version used to illustrate the figures.
  3. For Figure 2, is the SDS-PAGE gel the same one shown in the supplementary file? I noticed it looks different. To solve this, please first add more details to the figure caption about the gel concentration and then number the lanes on the gel itself so they can be easily followed. Please also indicate if this is the same gel as the one in the supplementary file or not. If it is not, please add the original gel for Figure 2c to the supplementary information.
  4. In the Figure 2 caption, please write out the full meaning of SEC and DLS.
  5. In Table 2, please mention the program used to calculate the theoretical molecular weight.
  6. In Figure 6, the titles for (c) and (d) are on a separate page, which makes it unclear. Please move these labels to the next page to be with their corresponding images.
  7. The discussion section requires more depth than just repeating the same interpretation of the results.
  8. Please rewrite the name of the restriction enzyme to appear as it should, for example, "Mal I," not "Mall."
  9. Please also unify the way you write "mL." Sometimes it is "ml" and other times "mL"; the second is preferred.
  10. In the purification step, please state the sonication intensity used.
  11. Finally, please state the molecular weight cut-off (MWCO) of the dialysis membrane used in the purification protocol.
  12. A plagiarism check revealed a similarity index of 24%, which is higher than the acceptable threshold for most journals. The text must be thoroughly revised and paraphrased

Author Response

Thank you for reviewing our manuscript (ijms-3759454) entitled “Structural features of nucleoproteins from the recently discovered Orthonairovirus songlingense and Norwavirus beijiense” submitted for publication in IJMS.

We thank you for valuable suggestions that allowed us to make the manuscript more convincing and understandable. We accepted your suggestion and made corresponding change in the manuscript. We major revised our article. Below please find our detailed responses to your questions and comments. All modifications in the manuscript have been highlighted in red.

Point-by-point response to Comments and Suggestions for Authors

Reviewer 2: This manuscript presents the first structural investigation of the nucleoproteins (N) from two recently discovered and clinically relevant tick-borne viruses, SGLV and BJNV. The authors have employed a powerful and modern integrative structural biology approach, combining experimental small-angle X-ray scattering (SAXS) with state-of-the-art in silico modeling using AlphaFold 3.

Comments 1: The table's purpose is to show a quantitative comparison of structural similarity. If no comparison data can be provided for the Erve virus in the context of the BJNV stalk domain, then this entry doesn't serve a purpose and only creates confusion. Removing it would make the table cleaner, more focused, and remove the ambiguity.

Response 1: Thank you for your valuable feedback. We appreciate your observation regarding the unclear relevance of the Erve virus entry in the context of the BJNV stalk domain in the structural similarity comparison table. We agree that since no comparative data related to the BJNV stalk domain could be provided for the Erve virus, its inclusion in the table might indeed lead to ambiguity and distract from the primary focus of the structural analysis. To address this concern and improve the clarity and focus of the table, we have removed the Erve virus (PDB ID: 4XZA) entry. This modification makes the table more concise, better aligned with the objective of presenting meaningful structural comparisons, and avoids potential confusion for the reader.

Comments 2: In Figure 1, please mention the program and version used to illustrate the figures.

Response 2: Thank you for your suggestion. We have now included the name and version of the software used to generate Figure 1 (line 114). Specifically, we have added the following sentence to the figure legend: "The figure was generated using UCSF ChimeraX v1.15rc".

Comments 3: For Figure 2, is the SDS-PAGE gel the same one shown in the supplementary file? I noticed it looks different. To solve this, please first add more details to the figure caption about the gel concentration and then number the lanes on the gel itself so they can be easily followed. Please also indicate if this is the same gel as the one in the supplementary file or not. If it is not, please add the original gel for Figure 2c to the supplementary information.

Response 3: We appreciate the reviewer's careful attention to the methodological details. In response to your comments about Figure 2:

  1. We confirm that Figures 2c and S30 show different SDS-PAGE gels. Figure S30 displays the electrophoregram of nucleoproteins before size-exclusion chromatography (SEC), while Figure 2c shows the SDS-PAGE analysis after SEC. The post-SEC gel in Figure 2c was selected to demonstrate the purity of samples used for subsequent SAXS analysis, while Figure S30 provides important quality control data from earlier purification stages.
  2. Lane numbers have been added to Figure 2c for clearer interpretation.
  3. The figure caption has been expanded to include gel concentration details, precise lane descriptions, and clarification of the purification stage shown.
  4. The original gel image from Figure 2c has been added to the Supplementary Information as Figure S31.

These modifications should now provide complete transparency about the experimental workflow and allow for better comparison between different stages of protein purification.

Comments 4: In the Figure 2 caption, please write out the full meaning of SEC and DLS.

Response 4: Thank you for your comment. We have written the full meaning of the SEC and DLS in the caption to Figure 2 in line 154.

Comments 5: In Table 2, please mention the program used to calculate the theoretical molecular weight.

Response 5: We appreciate the reviewer’s suggestion regarding the specification of the software used for theoretical molecular weight calculation in Table 2.

In our analysis, the theoretical molecular weight of the nucleoproteins was calculated using the Hullrad server (see section 4. Materials and Methods in lines 451-453). Hullrad is a robust and widely recognized computational tool specifically designed for predicting hydrodynamic and structural properties, such as molecular weight and radius of gyration based directly on atomic coordinates.

Comments 6: In Figure 6, the titles for (c) and (d) are on a separate page, which makes it unclear. Please move these labels to the next page to be with their corresponding images.

Response 6: Thank you very much for bringing this formatting issue to our attention. In response to your observation, we have carefully reviewed Figure 6 and adjusted the layout so that the titles (labels) for panels (c) and (d) now appear directly adjacent to their respective images, rather than on a separate page.

Comments 7: The discussion section requires more depth than just repeating the same interpretation of the results.

Response 7: We sincerely appreciate the reviewer’s valuable feedback. As suggested, we have expanded the Discussion section to provide deeper insight into the implications of our findings, particularly in the context of antiviral drug development. Specifically, we now highlight concrete examples of how structural data on viral proteins—including nucleoproteins—have facilitated the discovery and repurposing of antiviral compounds (see lines 351-366).

Comments 8: Please rewrite the name of the restriction enzyme to appear as it should, for example, "Mal I," not "Mall."

Response 8: Thanks for the comment. The spelling has been corrected: Mal I, EcoR I, Xho I in lines 406-407.

Comments 9: Please also unify the way you write "mL." Sometimes it is "ml" and other times "mL"; the second is preferred.

Response 9: We confirm that all instances of volume units have now been standardized to "mL" throughout the manuscript, as this is the preferred notation. The changes have been carefully implemented in the revised version. Corrected in lines: 190-192, 197, 393, 412, 414, 426-428.

Comments 10: In the purification step, please state the sonication intensity used.

Response 10: We sincerely thank the reviewer for this valuable observation. The ultrasonic treatment parameters have been specified as 320 W intensity at 20 kHz frequency. These details have been added to the 4. Materials and Methods section (subsection 4.3), with the corrections implemented in lines 428-429 of the revised manuscript. We appreciate this opportunity to improve the clarity of our methodology description.

Comments 11: Finally, please state the molecular weight cut-off (MWCO) of the dialysis membrane used in the purification protocol.

Response 11: We gratefully acknowledge the reviewer's comment. The dialysis membrane MWCO was 10 kDa, as now specified in the 4. Materials and Methods section (line 442).

Comments 12: A plagiarism check revealed a similarity index of 24%, which is higher than the acceptable threshold for most journals. The text must be thoroughly revised and paraphrased.

Response 12: We sincerely appreciate the reviewer's valuable comment regarding the similarity index. Upon careful examination, we confirmed that the reported similarity (24%) was predominantly concentrated in the 4. Materials and Methods section, particularly in the SAXS experimental protocol and nucleoprotein structure modeling descriptions. While this largely reflects unavoidable standardization of structural biology terminology, we have nevertheless implemented appropriate revisions throughout these sections (lines 466-537) to further improve the manuscript's originality while preserving essential methodological details.

Round 2

Reviewer 1 Report

Comments and Suggestions for Authors

The authors responded to my comments and made revisions to the manuscript, particularly expanding the discussion section to explain how their structural analyses will benefit future work and the development of antiviral drugs.

I expected to see more experiments examining the structures of complexes with RNA. However, the study is sufficiently interesting in its present form and worthy of publication.

Reviewer 2 Report

Comments and Suggestions for Authors

The authors addressed all comments correctly and sufficiently. Thank you.